# The Influence of Tourism Development Strategies on the Attractiveness of Mountainous Destinations: A Case Study of the Aures Mountains in Algeria

**Salah Zeraib [1], Yacine Kouba [2,*] and Belkacem Berghout [3]**

[1] Departement of Geography and Regional Planning, Institute of Earth Science and Universe, University of Batna 2, 53 Road of Constantine, Batna 05078, Algeria

[2] Department of Geography and Regional Planning, University of Larbi Ben M'hidi, Oum el Bouaghi 04000, Algeria

[3] Institute of Applied Sciences and Technologies, University of Skikda, BP 26, Road of El Hadaiek-Skikda, Skikda 21000, Algeria

* Correspondence: yacine.kouba@univ-oeb.dz

**Abstract:** Tourism development strategies play a crucial role in tourism development. However, the reaction of the former to the needs of visitors and its effect on attractiveness is essential, especially in mountainous destinations. This study evaluates the impact of tourism development strategies on the attractiveness of mountain destinations. The study relied on appropriate elements derived from the literature. The study was conducted in three tourist sites in the Aures Mountains, and the sample included 468 visitors. The results showed that the destination's attractiveness depends mainly on local factors such as nature, monuments, traditional food, and apple purchase, in addition to the quality of the price, which received the satisfaction of the majority of visitors. In turn, visitors were dissatisfied with the services assigned to tourism development strategies, such as accommodation, entertainment, communications, and transportation. The results also showed that the return to the destination is affected by nature and determined by several factors such as age, gender, use of a specific vehicle, and proximity. Therefore, the destination's attractiveness is not based on the elements assigned to tourism strategies; this indicates the gap in local potential and tourism development.

**Keywords:** mountains; tourism; tourism development strategies; attractiveness; attraction factors; destination

## 1. Introduction

Given the geographical and economic aspects, tourism is vital in establishing functional links between the many productive areas characterized by tourism activities [1–4]. It represents an appropriate activity to achieve development in mountainous regions [5,6]. The importance of mountain tourism has led governments and relevant authorities to develop particular tourism development strategies for these destinations [6–8]. These strategies aim to enhance local communities' awareness, mountains' development, degradation reduction, and deepening knowledge about mountainous regions [9]. Other objectives, such as the development of the economic interests of local communities [9], can also be added. Thus, tourism development strategies in mountainous areas represent a tool that provides the necessary measures to achieve sustainable development based on their exceptional characteristics [9,10]. Therefore, national planning is crucial in managing and developing infrastructure in mountain destinations to strengthen links between attractions, facilities, services, and tourism activities [8,11], especially in developing countries [12,13]. In many countries, mountain tourism has evolved from local attractions to internationally recognized destinations [14], becoming one of the most popular tourist destinations [5]. Several studies on the relationship between tourism development and destination attractiveness have been conducted, using various indicators such as length of stay and economic gains.

The majority of them demonstrated that good levels of tourism development improved the quality of service and the attractiveness of the destination [15,16].

Mountain tourism in Algeria faces a significant lack of infrastructure and adequate facilities to receive tourists [17] due to the lack of a clear strategy for developing mountainous areas [17]. Until 2005, all plans [tripartite (1967/1969), quadrilateral (1970/1977), and pentagonal (2001/2005)] aimed at building tourist expansion zones (TEZ) and attracting foreign investment for tourism development in coastal and desert areas [18]. However, these initiatives have not been able to support tourism development due to political turmoil and administrative bureaucracy [17]. The Master Plan for Tourism Development (MPTD) was made up in 2008 to address the previous gaps. The state adopted it as a strategic reference for the tourism development policy in Algeria during the period 2008–2030. In forming the MPTD, the government relied on trends that promote the development of distinct regions through rationalizing investment and development following new global trends. Six major goals were identified: promotion of the Algerian destination, establishment of seven tourism poles to serve as a real front and symbol of Algerian tourism's emergence, facilitation of public-private partnerships, encouragement and support of national and international investment in the sector, and development of qualifications and skills. Finally, the Tourism Quality Plan (TQP) has been published to encourage the use of information and communication technology (ICT) and the development of links between key tourism sectors such as accommodation, restaurants, parks, resorts, and cultural services, as well as business lines directly related to tourism such as transportation and security services [19]. Mountainous areas are marginalized in the MPTD because they are not proposed as distinct tourist poles, which contradicts the plan's essential sustainability principles. The marginalization of mountain areas discourages the development of mountainous tourist products that play a crucial role in shaping the demand and behavioral intentions of the tourist and return decision recommendation [20,21].

This paper aims to investigate the impact of tourism development strategies on the attractiveness of mountain destinations. Several elements were examined to understand the factors of destination attractiveness and determine elements affecting the repeat of visit (return). The purpose is to understand the role of the assigned elements of tourism development strategies/facilities and services in the attractiveness of the mountain destination. The importance of this study lies in the fact that it is the first to discuss the issue of tourism in the Aures Mountains, using the satisfaction indicator to measure the quality of service and its impact on the destination's attractiveness.

## 2. Literature Review

Recently, mountain regions have received growing attention from researchers regarding issues of mountain tourism [4,22–24] and have become an interesting area of controversy with issues of planning, tourism development, and recreation [25]. They highlighted tourism as a promising strategy to provide mountainous communities with alternative options to earn a living [21]. Tourists represent an essential component of mountain tourism activity and development through their demands that play important roles in developing tourism products [26]. Weidenfeld et al. and Leask et al. [27,28] assert that demand generation in shaping destinations' attractiveness is based on nature tourism, which is based on attractions and values. Moreover, Lew [29] acknowledges that attractions are the main element of tourism development. Additionally, Grandpré [30] explains that attraction factors are the natural and cultural resources in the regions that can contribute to tourism products.

Significantly, the difference in the concept of attraction factors in tourism literature led researchers to adopt multi-dimensional models. One of the first attempts to visualize the tourist attraction model was made by MacCannel [31], who concluded that destination attractiveness is a combination of three components, the tourist, the scene, and the sign. These represent part of the information about only the scene. Gunn [32] formed an attraction structure consisting of three concentric circles, the inner circle representing the nucleus,

the most important component of the attraction factors. The second essential component is the dielectric zone, which secures the core and acts as a tourism supervisor. The third component is the area around the attraction, which contains tourist services. Despite the importance of the previous models in explaining the factors of attraction, they focus more on tangible elements. In light of that, Vittersø et al. [33] developed a model in which he demonstrates the importance of subjective meaning, including the symbolic and emotional values that tourists attribute to attractions. Adopting these approaches on the principle of a major attraction factor may affect the components of the service, which are essential for the operation of any tourist destination [14,34]. This does not enhance the destination's attractiveness and the development of intensive tourism activity even with the availability of a major attraction factor [29,35].

In this context, Yang [36] demonstrates the extent of the tourist movement's influence by the structure and dimensions of touristic destinations' integration of the principal attraction factors and the touristic services that may not always align with government-supported regulations. Therefore, this multi-faceted nature of tourist destinations represents a significant challenge for tourism development strategies in matching tourism resources and tourist attractions with tourism motives and preferences [35,37]. Thus, despite their centrality, the attractions are part of a complex tourism network within the destination, which plays a crucial role in structuring the tourist offer system [30,38]. As for the attractiveness in this approach (the approach to attractiveness focuses instead on the tourist destination), it is considered a group of functions of the tourist's perception of the destination's ability to meet their needs and provide personal benefits [38,39]. Scientists have widely determined the attractiveness of a tourist destination. Accordingly, Mayo and Jarvis [40] define the concept of destination attractiveness as a combination of the relative importance of individual benefits and the perceived ability of a destination to deliver unique benefits. This ability is enhanced by specific attributes that make up a destination, such as attractions, infrastructure, or services and the people who provide them. Leiper [41] and Vengesayi et al. [42] consider that destination attractiveness represents visitors' opinions about a destination's abilities to meet their needs and goals. Lue et al. [43] state that it is something individuals recognize that influences their decisions about leisure travel. Kim and Perdue [44] see that a tourist destination's attractiveness is a set of tourist facilities and services.

Furthermore, the attractiveness of a destination, in terms of the spatial dimension, constitutes the geographic areas that offer a mix of tourism products and services [45]. In this context, the allure of mountainous regions is based on fresh air, natural landscapes, wildlife, scenic beauty, rich cultural heritage, and recreational opportunities [4,6,16,34]. Relatively, Wang [46] confirms that natural components comprise major factors in visiting mountainous destinations. Then, Debarbieux et al. [47] add that comfortable factors such as the calm environment encourage visits to mountainous areas. However, Needham et al. [48] argue that the beauty of nature and the infrastructures compose the main factors for the attractiveness of mountainous destinations. In addition, said attractiveness can be mainly related to some elements, such as the availability of facilities and quality of services [49]. These results indicate the importance of integrating the main attractions and the elements assigned to tourism development strategies, such as facilities, services, and activities, in the destination's attractiveness and the experience's quality.

Several researchers studied the effects of the experience and perception on satisfaction and behavioral intention to understand tourism development's relationship with the experience's quality [16]. Significantly, Chen and Chen [50] discussed the method of decision-making design with a service product. They found that the relationship between decision-making and service products represents a complex system; consumer value perception affects satisfaction and behavioral intention. Moreover, Zeng and Yi Man Li [51] found that empirical value influences behavioral intentions via leisure and ecotourism satisfaction. In light of that, emotional responses are important in measuring satisfaction with the quality of tourism products and their impact on behavioral intention/decision-making to recommend and return to the destination. Thus, paying attention to the relationship

between development and tourism and its contribution to the general satisfaction of tourists helps to develop destinations and enhance their attractiveness. Therefore, using the visitor satisfaction and behavioral intention index is important in analyzing and understanding the impact of tourist attractions to determine the relationship between tourism development strategies and the attractiveness of destinations.

## 3. Background

The Aures Mountains are located in the eastern part of the Saharan Atlas, which separates Algeria's northern and desert regions (Figure 1). The region enjoys a variety of landscapes and covers an area of about 9000 square kilometers, with an altitude of 2326 m (the top of the mountain is Chelia). It has a rich historical background [52] and witnessed many civilizations, most notably the Roman civilization. One of its most important landmarks is the ancient city of Timgad (Figure 1), considered one of Algeria's most important archaeological sites [53]. Among the most important things that distinguish these mountains are the great diversity of cultures and landscapes, the beautiful oases, villages with different urban patterns, and the high biodiversity harbored by the Aleppo pine and Atlas Cedar forests [53].

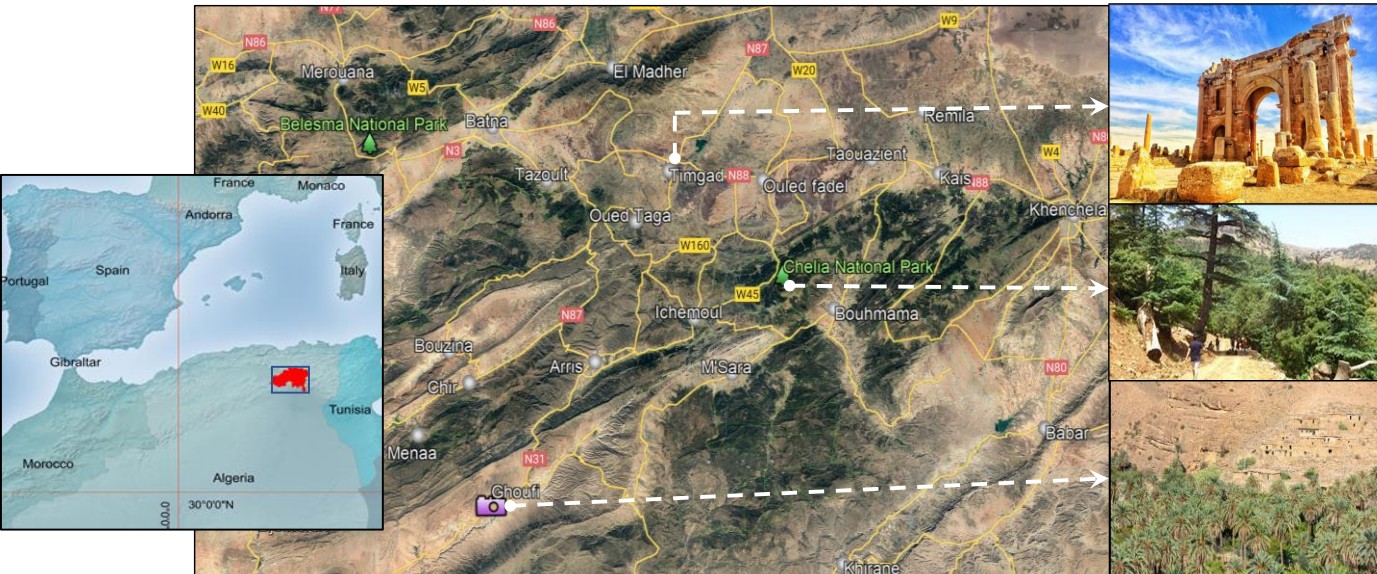

**Figure 1.** The left panel represents the location of the Aures region in Algeria. The middle panel includes google earth photography of the study area, showing the main urban areas, road network, and the two national parks. The right panel encompasses three photographs showing, from top to bottom, Timgad ruins, Cedar forest of Chelia national park, and Ghoufi oasis, respectively.

Moreover, the Aures Mountains represent the most important apple-producing areas in Algeria. Considering this potential, it may allow it to be an important tourist destination. For example, during the spread of COVID-19 in 2020, the Aures Mountains attracted large numbers of visitors, contributing to the revival of many traditional dishes (such as Mardoma, Al-Zirawi, and Al-Rafis) and some traditional industries (textiles, pottery), which encouraged the marketing of apple products. However, the lack of accommodation (Table 1) and other tourist facilities prompted most visitors to spend a short period in the tourist sites, not exceeding a few hours. Thus, this may not encourage the destination's attractiveness, and it indicates the existence of a gap between tourism development strategies and local capabilities. Therefore, visitor evaluations are important to know the factors affecting the destination's attractiveness to determine the role of tourism development strategies.

**Table 1.** Evolution of bed capacity by type of tourist product in Algeria in general and the Aures Mountains in particular, according to the official Algerian statistic office.

|  | 2010 | 2012 | 2013 | 2014 | 2015 | 2016 | 2017 | 2018 | 2019 | 2020 | 2021 |
|---|---|---|---|---|---|---|---|---|---|---|---|
| Urban | 52,085 | 54,186 | 55,988 | 61,012 | 62,479 | 69,135 | 69,861 | 74,712 | 80,470 | 81,863 | 85,577 |
| Seaside | 31,322 | 29,886 | 29,886 | 27,962 | 30,380 | 32,200 | 31,326 | 32,581 | 32,926 | 32,971 | 33,588 |
| Saharan | 3770 | 5954 | 6058 | 4547 | 3636 | 4912 | 4928 | 5477 | 5895 | 6299 | 6620 |
| Thermal | 4111 | 5467 | 5467 | 4259 | 3866 | 4202 | 4266 | 4502 | 4502 | 4598 | 4598 |
| Mountainous | 1089 | 1405 | 1405 | 1825 | 1883 | 1883 | 1883 | 1883 | 1883 | 1883 | 1883 |
| Aures Mountains | 32 | 32 | 32 | 32 | 32 | 32 | 32 | 32 | 50 | 50 | 50 |

## 4. Materials and Methods

### 4.1. Data Collection

A questionnaire-based survey was used for data collection. After a bibliographical review of questionnaire construction, the questionnaire was divided into two sections. The first one comprised questions about visitors' origins, demographic information (age, gender), tourist destination, the purpose of visit, mode of travel, source information, length of stay, and visit frequency. The second section included questions about visitors' satisfaction with the visited site and the level of tourism facilities and services, such as accommodation, communication, pricing, transport, entertainment, and roads condition. Each respondent's answer was rated on a five-point (1 to 5) Likert scale (very dissatisfied, dissatisfied, neutral, satisfied, very satisfied).

A preliminary pilot test of the questionnaire was conducted by distributing it to a diverse group of people with varying employment, life situation, hobbies, and attitudes toward mountain activities. Their feedback and comments were used to make revisions and adjustments. Five hundred questionnaires were re-distributed to a random sample of visitors who agreed to participate in the study between 18 December 2021, and 4 January 2022 (i.e., during the winter holiday of schools and universities). The questionnaires were distributed at the most prominent tourist destinations of the Aures Mountains, namely the Ghoufi oasis and mount Chelia, which were classified as national nature reserves in 2006, and the Timgad ruins listed as a world cultural heritage site by UNESCO in 1982. After reviewing the gathered questionnaires, 32 were eliminated due to insufficient answers, and the rest (468 questionnaires) were used for the statistical analysis.

### 4.2. Data Analysis

The questionnaire data were analyzed with R 4.2.0 (R Foundation for Statistical Computing, Vienna, Austria). First, to analyze tourist satisfaction, factor analysis was used; the data were tested to ensure they met the requirements for factor analysis using the Kaiser-Meyer-Olkin measure of Sampling Adequacy and Bartlett's test of Sphericity. Kaiser's Criterion (eigenvalue > 1) and Screen Test were used to determine the number of factors to be retained, and the Varimax method was used for factor rotation. Cronbach's alpha was used to verify the internal consistency reliability of the extracted factors.

Secondly, the most important factors influencing visit frequency were determined. A linear model was fitted using the following variables as predictors: visitor gender (males, females), visitor age (years), the distance between the touristic site and the visitor's home town, mode of transport (own vehicle, public transport), the purpose of visit (buying apples, visiting ruins, enjoying nature). Visit frequency was used as a dependent variable in the model. Model selection procedure based on the corrected Akaike information criterion (AICc) was used to select the best predictors of visit frequency. Model residuals were inspected to ensure homoscedasticity and normality. All predictors and response variables were standardized using Z-score to interpret parameter estimates on a similar scale.

## 5. Results

### 5.1. Sample Characteristics

The analysis showed that 16.03% of the respondents visited the destination for the first time, 18.16% for the second time, 27.83% for the third time, and 37.98% repeated their visit more than four times (Table 2). The responses to the purpose of the visit revealed that the reasons for enjoying nature (nature-based tourists) represented the highest percentage with 81.84%, followed by apple buyers (visitors whose main reason for coming is to buy apples) with 12.18%, and finally 5.98% to ruins visitors (visitors whose main reason for coming is to visit ruins). In response to the length of stay, 91.24% of participants preferred to stay less than one day, 6.20% to two days, and 2.56% preferred to spend more than three days. When asked where they heard about the site, 16.45% said they heard about it from friends, 82.69% via social media, and 0.85% of tourists from official media (TV). Due to the difficulty of accessing the Aures Mountains using public transport, 86.11% of the respondents preferred to come with their vehicles, while only 13.89% of the surveyed visitors used public transport. The survey revealed male (70%) dominance over females (30%), and the average age of the visitors ranged between 25 and 37 years.

**Table 2.** Travel information of the respondents (*n* = 468).

| Purpose of Visit | Frequency | % | Length of Stay | Frequency | % |
|---|---|---|---|---|---|
| Enjoying nature | 383 | 81.84 | One day | 427 | 91.24 |
| Visiting ruins | 28 | 5.98 | Two days | 29 | 6.20 |
| Buying apples | 57 | 12.18 | Three days or more | 12 | 2.56 |
| Origin of tourist | Frequency | % | Visit frequency | Frequency | % |
| Local | 358 | 76.5 | First time | 75 | 16.03 |
| Near provinces | 55 | 11.75 | Second time | 85 | 18.16 |
| Far provinces | 42 | 8.97 | Third time | 62 | 13.25 |
| Foreigners | 13 | 2.78 | Fourth time or more | 246 | 52.56 |
| Tourist destination | Frequency | % | Source information | Frequency | % |
| Chélia | 44 | 9.40 | Friends | 77 | 16.45 |
| Ghoufi | 386 | 82.48 | Social media | 387 | 82.69 |
| Timgad | 38 | 8.12 | Television | 4 | 0.85 |
| Mode of travel | Frequency | % | Sex | Frequency | % |
| Public transport | 65 | 13.89 | Males | 333 | 71.15 |
| Own vehicle | 403 | 86.11 | Females | 135 | 28.84 |

The majority of respondents (76.5%) came from provinces close to the destination (Biskra, Khenchela, and Batna, with a mean distance of 50 km), and 11.75% came from neighboring provinces (Oum el Bouaghi, Setif, and Constantine, with a mean distance of 150 km), 8.97% came from the rest of the country, and 2.78% were foreigners (Figure 2). The survey identified the Ghoufi oasis as the main tourist destination during the survey period as more than 82.48% of the surveyed visitors chose this destination, while Chélia mount and Timgad ruins accounted for 9.4% and 8.12% of the surveyed visitors, respectively. These proportions do not explain the importance of the tourist sites. Still, they are related mainly to the advantage of the site's climatic conditions; for example, the location of Ghoufi oasis attracts more visitors during winter and spring. Overall, these results indicate that most of the visitors are young, and many visit the site several times. Tourism in the region is domestic, a typical tourism feature in mountainous destinations. Nature plays an essential role in attracting visitors. Promotion through public media and public transport is very weak. The stay is very short, which does not encourage sustainable development in the region.

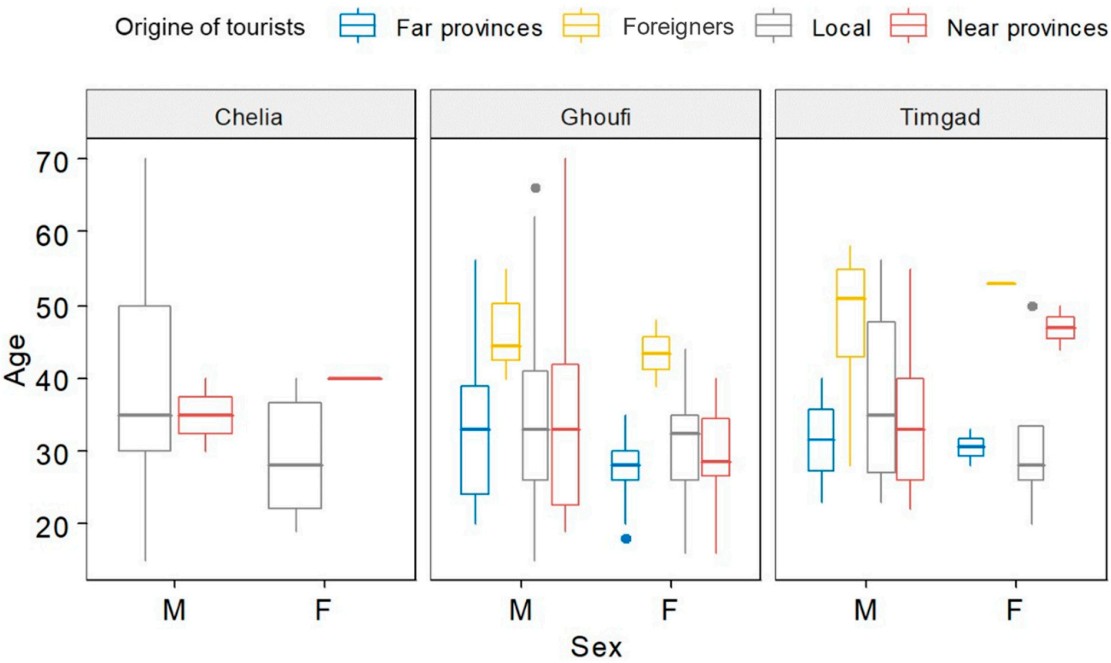

**Figure 2.** Respondents' ages as a function of their origin, sex, and tourist destination.

*5.2. Factors Affecting Tourist Satisfaction*

The Kaiser-Meyer-Olkin measure of sampling adequacy (KMO = 0.728) and Bartlett's test of Sphericity [X2(141) = 98.2, sig. 0.000] indicated that the sample size and the data were adequate for conducting factor analysis. A criterion for eigenvalues equal to or greater than 1.00 extracted two factors, explaining 53.11% of the total variance. The factors showed perfect Cörnbach's alpha levels, with a value of 0.823 for the first factor and 0.803 for the second factor (Table 3). The first factor (F1), with an eigenvalue of 4.972, explained 34.675% of the total variance, and the second factor (F2) explained 18.435% of the variance with an eigenvalue of 2.743. F1 represents tourism facilities and contains five items: entertainment, communication, transport, road condition, and accommodation. The five items are strongly associated with F1, as they have a correlation of 0.874, 0.716, 0.706, 0.695, and 0.718, respectively. However, these items tend to disagree based on their mean scores ≤ 2.75. F2 refers to the region's potentialities and encompasses pricing (loading = 0.523), site attractiveness (loading = 0.637), and traditional food (loading = 0.542). The item site attractiveness tends to strongly agree according to its mean score of 4.12, and the variables pricing and traditional food tend to agree according to their mean scores of 3.95 and 3.86, respectively.

**Table 3.** Results of principal factor analysis (factors structure and item loadings, along with factor' eigenvalues, the percentage of explained variance, and Cronbach's alpha). Descriptive statistics of variables (mean and standard deviation) are also included.

| Factor | Item | Mean | SD | Item Loading | Eigenvalues | Variance | Cronbach's Alpha |
|---|---|---|---|---|---|---|---|
| F1: Tourism facilities | | | | | 4.972 | 34.675 | 0.823 |
| | Entertainment | 2.02 | 0.23 | 0.874 | | | |
| | Communication | 2.14 | 0.45 | 0.716 | | | |
| | Transport | 2.12 | 0.75 | 0.706 | | | |
| | Roads condition | 2.75 | 0.46 | 0.695 | | | |
| | Accommodation | 2.17 | 0.68 | 0.718 | | | |
| F2: Region potentialities | | | | | 2.743 | 18.435 | 0.803 |
| | Pricing | 3.95 | 0.27 | 0.523 | | | |
| | Site attractiveness | 4.12 | 0.21 | 0.637 | | | |
| | Traditional food | 3.86 | 0.26 | 0.542 | | | |

*5.3. Factors Affecting Repeat of Visit*

Determining the factors that affect visitors' repeated visits helps to understand the strength of satisfaction and the behavioral intentions of visitors [54]. Figure 3 represents the dot-and-whisker plot of the regression model, in which the predictor estimates are presented as dots and their confidence intervals as whiskers. This figure reveals the mode of travel [own vehicle], the purpose of the visit [buying apples], gender [males], and age of visitors as factors that significantly positively influence repeat of visit. However, the distance from the Aures Mountains and the purpose of the visit [visiting ruins] significantly negatively affect the visit frequency. These results mean that visit frequency increases among those visitors (1) who use their cars for traveling to the site, (2) whose primary purpose is buying apples, (3) that are men, (4) that are adults, or (5) live in proximity or came from near cities. Nevertheless, those tourists whose purpose is visiting ruins or those that live far from the visited site tend to visit the site less frequently.

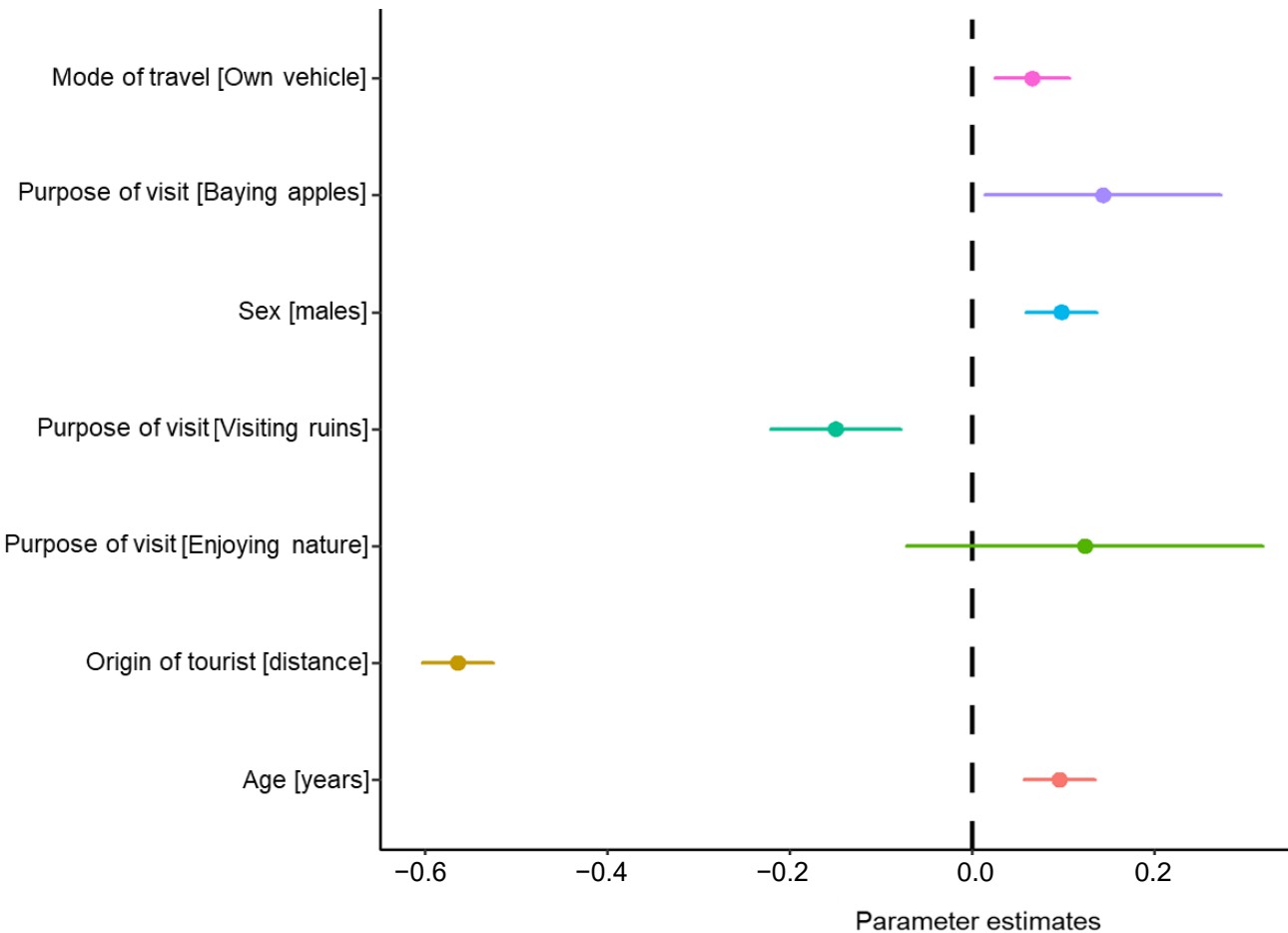

**Figure 3.** Model-averaged parameter estimates and 95% confidence intervals for the seven explanatory variables for the number of visits to the Aures Mountains.

## 6. Discussion

The findings of this research confirmed that tourism development strategies have not been able to achieve the necessary integrations to build the tourist structure of the destination and enhance its attractiveness. Similar findings were found in other regions of Algeria [17,18,55]. The results also showed a variation in destination attractiveness, with the site of Ghoufi accounting for 84% of the total visitors (Table 2). Located 45 km northeast of Biskra and 75 km south of Batna, Ghoufi is characterized by prominent natural and cultural resources, such as rock gorges and mild winter weather, ruins of local villages, palm oases, as well as traditional foods. This resource diversity indicates the importance of complementarities between natural and cultural elements in supporting the destination's attractiveness. These features may incentivize attracting visitors to this particular site more than others, which is in line with the literature that finds that the selected elements do not contribute the same values that determine attractiveness [56].

The results revealed two diverse factors affecting the destination's attractiveness (Table 2). The first group is not influential as it did not satisfy most visitors and encompasses tourist facilities and services such as accommodation, public transport, communications, road conditions, and entertainment. The second group, which affects the destination's attractiveness, is based mainly on the region's capabilities, such as nature, ruins, traditional food, cheap and high-quality apples, and affordable prices of traditional food (Table 3). This latter element is essential in supporting the destination's attractiveness and contributes significantly to visitors' satisfaction [57,58]. Traditional restaurants are considered the most important infrastructure facilities in the Aures to meet visitors' needs. These restaurants are widespread in the destination and accessible due to the relative improvement of the road

network due to the local development initiatives undertaken at the beginning of the first decade of the current century to overcome the region's isolation. This can be considered a positive indicator that partly reflects the development of road infrastructure investment and may enhance visitors' flow [59,60].

Local resources played crucial roles in strengthening the link between the Aures destination and its visitors. They gave it a favorable image by promoting it via social networks and recommending its visit (Table 2). It is a goal not achieved by government programs despite implementing the Master Plan for Tourism Development for 2008–2030, which considers promoting tourist sites a crucial element of the tourism development strategy. However, promoting the Aures Mountains as a tourist destination has yet to be accomplished within the national plans. Increasing the available information about nature and monuments classified globally, such as Timgad monuments, may improve the attractiveness and contribute to the promotion of the destination [61]. It appears illusive owing to the lack of tourism facilities and services that respond to visitor demands and the lowest quality standards.

The failure of tourism development strategies to achieve their goals led the destination to acquire a local/popular touristic mold. This is consistent with the majority of the literature that confirms the lack of services and facilities for tourism in developing countries and mountainous regions [21,62]. Therefore, this often leads visitors to spend a short while at the destination, not exceeding one day (Table 2). Despite this, the destination witnessed visitors' return, indicating that visitors did not care much about the tourist facilities and services that were negatively related to their satisfaction, in contrast to local resources such as nature and monuments, which were positively correlated with the attractiveness of the destination. Despite that, it had no apparent effect on the return of visitors; instead, the return of visitors to the Aures destination is determined by other elements, namely: (1) visitor gender and age (Figure 3); most visitors are males and young; (2) the use of a private car, which helps to reach the destination quickly; (3) visiting the area to buy high-quality and affordable apples, and (4) the proximity of the visitor's origin to the destination, which allows visitors not to spend a long time at the destination. These elements are not limited to a particular location but characterize all destination sites (except for the archaeological site of Timgad), where the farther away, the lowlier the odds of return (Figure 3). Previous studies suggested that the elements of the frequency of the visit are important in enhancing the attractiveness of the destination [63,64].

In an attempt to assess the impact of tourism development strategies in the context of the attractiveness of the destination of the Aures Mountains in Algeria, the survey was limited to one time period of the year. It is assumed that if it was conducted during different periods of the year, it might be more varied in terms of the diversity of the visitors' demographic characteristics, such as gender and age, in addition to the variety of visitors' origins and objectives of their visits. These components affect the attractiveness of mountainous touristic destinations [52,65]. The current study also lacks the views and perceptions of decision-makers and stakeholders that affect tourism development strategies and plans [56,66].

Our study shows the enormous gap between the local potential of the region and the insufficient contribution of tourism development strategy programs to the destination's attractiveness. This poses many challenges to the authorities concerned with tourism development strategies to enhance and upgrade the destination's attractiveness and raises controversy over the tourism situation in the Aures Mountains and other similar mountains in Algeria. Overall, the current study falls within the tourism development literature. It adds its results from an academic point of view, which provides a conceptual framework that contributes to progress in developing tourism development strategies' impact on the attractiveness of mountain destinations. From a practical point of view, it highlights some of the challenges faced by the attractiveness of mountainous regions. Its results may benefit a wide segment of society, decision-makers and stakeholders, tour operators, travel agencies, and tourists.

### 7. Suggestions for Sustainable Tourism Development in the Aures Mountains

Although tourism may play an essential economic role, and its growth can help develop related sectors, tourism in the Aures Mountains is still far from the goals of state-controlled programs for sustainable tourism development. Hence, to achieve sustainable tourism development in this mountainous area, the state should pay attention to tourist satisfaction and invest in the development of basic tourism facilities to create a tourist destination that can be a source of competitive advantage. The state should focus primarily on investing in tourist facilities because they represent the weak link in the tourism products in the Aures Mountains. Accordingly, the entertainment, accommodation, communication, roads condition, and transport facilities must be improved, and the minimum standard must be respected. The poor level of services may encourage arbitrary tourism, so the sites with high biodiversity and archaeological sites must be preserved by including the affected and at-risk destinations within the national reserves.

The work of local associations working in the management of tourism must support the education of visitors and residents about the importance of natural and cultural monuments in stimulating tourism in the region. Traditional food should also be encouraged by establishing schools specializing in traditional cooking to preserve it as the only local cultural component contributing to regional tourism. The distinctive agricultural products of the region should also be encouraged because they attract visitors' attention, provide a competitive advantage, and promote agricultural tourism in the area. Therefore, the tourism departments must provide the necessary information about the region's potential to market the destination and set policies to achieve sustainable development, maintain social and economic balances, and improve local communities' quality of life.

**Author Contributions:** Conceptualization: S.Z.; Method and data analysis: Y.K.; Revision and editing: Y.K., S.Z. and B.B. All authors have read and agreed to the published version of the manuscript.

**Funding:** This research received no external funding.

**Institutional Review Board Statement:** The study was conducted according to the guidelines of the Declaration of Helsinki and approved by the Ethics Committee of the University of Batna 2.

**Informed Consent Statement:** Informed consent was obtained from all subjects involved in the study.

**Data Availability Statement:** Available on request to the corresponding author.

**Acknowledgments:** We are thankful to the persons who helped us to carry out this work. The assistance of two anonymous referees in the improvement of the manuscript with their constructive comments is greatly acknowledged.

**Conflicts of Interest:** The authors declare no conflict of interest.

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
