# Peer review of "The Influence of Tourism Development Strategies on the Attractiveness of Mountainous Destinations: A Case Study of the Aures Mountains in Algeria"

_sustainability, doi:10.3390/su142013045_

Round 1

Reviewer 1 Report

The paper is significant not only for the area examined, but for the whole of the Algerian tourism, which contains numerous favourable attractions and amenities for tourism.

I suggest to the authors change of the following details:: 

Shorter version of chapter 1. Introduction, including fewer references.

It is necessary to strictly appoint and emphasize all of Algerian tourism strategies, plans, master plan etc. which you have analysed in your research. Furthermore, you need to emphasize main goals and directions of the newest strategy, especially those which could affect Algerian mountain regions and tourism development. Your article tittle suggests it!

Chapter 2. Literature review. You try to avoid unnecessary repeating. If I may advise you, try to focus on literature which considered impacts of tourism strategies on tourist attractiveness of certain mountain regions.

As the geographers, you have to arrange at least two geographical thematic maps in chapter 3. Background. The first map could explain geographical position and location of Aures Mountains in Algeria to larger groups of potential readers and users (not only those one inside geographical community) of your article. The second map of Aures mountains tourism region, and it has to contain main tourism resorts, roads, protected areas etc.

I recommend one table which it includes accommodations facilities and capacity according to official Algerian statistic office. That table must contain data about the number of tourist beds by main Algerian tourist regions. It is necessary to compare the significance of Aures Mountains in national tourism.

Try to avoid personal and private attitudes such as “… The Aures mountains are among the most famous mountains in Algeria and North Africa.”

In the subchapter 5.1., it is obligatory to exactly appoint far provinces and nearby provinces (names and average distance in kilometres)

In the legend of figure 1.: Foreigners, not Foreingers

In the table 1.: Buying apples, not Baying apples. However, I strongly doubt that apples buyers are tourists at the same time. So, I suggest in the subchapter 5.1., larger description and explanation about these groups (nature-based tourists, visitors of ruins, and apples buyers).

Chapter 6. Discussion. The findings of this research confirmed that tourism development strategies have not been able to achieve the necessary integrations to build the tourist structure of the destination and enhance its attractiveness. I cannot agree with this statement due to two issues.

Firstly, the questionnaire doesn’t investigate attitudes of tourists about implementation of strategy in their destination. Secondly, every tourism development strategy is only a document and nothing more than it. Tourism development in a certain region deeply depends on the entrepreneur’s environment, well-educated tourist staff, security etc.

I hope that my advice and recommendations will help.

Thank you

Kind regards

Author Response

Reviewer #1

Comment 1: Shorter version of chapter 1. Introduction, including fewer references.

Response 1: We have shortened the introduction section and reduced the number of references as suggested by the reviewer (see Introduction section.

Comment 2: It is necessary to strictly appoint and emphasize all of Algerian tourism strategies, plans, master plan etc. which you have analysed in your research. Furthermore, you need to emphasize main goals and directions of the newest strategy, especially those which could affect Algerian mountain regions and tourism development. Your article tittle suggests it!

Response 2: We are grateful to reviewer 1 for this comment. We have changed the introduction according to the reviewer's comment, which has  improved it greatly (see introduction section)

Comment 3: Chapter 2. Literature review. You try to avoid unnecessary repeating. If I may advise you, try to focus on literature which considered impacts of tourism strategies on tourist attractiveness of certain mountain regions.

Response 3: We have trained to remove the repeated ideas in the literature review as suggested by the reviewer. We are grateful to the reviewer for his advice. Still, our objective via this section is to highlight the relationship between the factors related to tourism development strategies ( accommodation, services, infrastructure) and destination attractiveness (see literature review).

Comment 4: As the geographers, you have to arrange at least two geographical thematic maps in chapter 3. Background. The first map could explain geographical position and location of Aures Mountains in Algeria to larger groups of potential readers and users (not only those one inside geographical community) of your article. The second map of Aures mountains tourism region, and it has to contain main tourism resorts, roads, protected areas etc.

Response 4: As suggested by the reviewer, we have added a new figure (Figure 1) that shows the location of the Aures region in Algeria as well as the main urban areas, the road network, and the protected areas in the region, together with three photographs of the surveyed sites.

Comment 5: I recommend one table which it includes accommodations facilities and capacity according to official Algerian statistic office. That table must contain data about the number of tourist beds by main Algerian tourist regions. It is necessary to compare the significance of Aures Mountains in national tourism.

Response 5: We are thankful to the reviewer for this comment. We have added a new table (Table 1) that shows the evolution of the number of beds per tourism type in Algeria, and the same statistics for the region of the Aures have also been added.

Comment 6: Try to avoid personal and private attitudes such as "… The Aures mountains are among the most famous mountains in Algeria and North Africa."

Response 6: This sentence has been removed in the new version of the MS (see L162).

Comment 7: In the subchapter 5.1., it is obligatory to exactly appoint far provinces and nearby provinces (names and average distance in kilometres).

Response 7: As suggested by the reviewer, we have added the names and average distance for closed and near provinces (see L247-251).

Comment 8: In the legend of figure 1.: Foreigners, not Foreingers.

Response 8: The mistake has been rectified.

Comment 9: In the table 1.: Buying apples, not Baying apples. However, I strongly doubt that apples buyers are tourists at the same time. So, I suggest in the subchapter 5.1., larger description and explanation about these groups (nature-based tourists, visitors of ruins, and apples buyers).

Response 9: The word has been corrected (see Table 2). We confirm that people (mainly from closed provinces) came to the region to buy apples and profit from the occasion to visit the archeological or natural sites. We have explained each tourist category in the text as suggested by the reviewer (see 235-239).

Comment 10: Chapter 6. Discussion. The findings of this research confirmed that tourism development strategies have not been able to achieve the necessary integrations to build the tourist structure of the destination and enhance its attractiveness. I cannot agree with this statement due to two issues.

Firstly, the questionnaire doesn't investigate attitudes of tourists about implementation of strategy in their destination. Secondly, every tourism development strategy is only a document and nothing more than it. Tourism development in a certain region deeply depends on the entrepreneur's environment, well-educated tourist staff, security etc.

Response 10: We are so grateful to reviewer 1 for this interesting comment, but with this sentence, we mean that the authorities responsible for the implementation of tourism development strategies failed to develop tourism infrastructure that could enhance destination attractiveness.

Reviewer 2 Report

Dear authors, thank you very much for the opportunity given to me to read this manuscript.

The topic - tourism development in mountainous areas - remains as one of much important in tourism studies and clearly points to critical issues in tourism destination management.

The reading of the manuscript is fluid and pleasant. 

Methodologically I find no critical points to address. Standard procedures have been followed and the analyses are consistent with data findings.

Minor issues refer to: (1) deletion of all traces of personal writing (e.g. "we") (2) the study's limitations, and (3) contributions to theory. It would be good a critical reflection on this. For instance, the sample included only visitors, and the majority was from nearby locations. Would the same results apply if respondents were mainly international or women? Can the destination, if interested, target these groups of visitors? Also you could point to potential development in tourism literature focusing on the topic you selected. Have you had some insights regarding this particular aspect? In contrast, managerial suggestions and recommendations are well done.

Author Response

Reviewer #2

Comment 1: Methodologically I find no critical points to address. Standard procedures have been followed and the analyses are consistent with data findings.

Response 1: We are so grateful to reviewer 2 for this comment.

Minor issues refer to: (1) deletion of all traces of personal writing (e.g. "we").

Comment 2: the study's limitations, and contributions to theory. It would be good a critical reflection on this. For instance, the sample included only visitors, and the majority was from nearby locations. Would the same results apply if respondents were mainly international or women? Can the destination, if interested, target these groups of visitors? Also you could point to potential development in tourism literature focusing on the topic you selected. Have you had some insights regarding this particular aspect?

Response 2: We are grateful to reviewer 2 for this comment which increased the MS greatly. Based on the suggestions made by the reviewer, we have added two paragraphs in the new version of the MS; the first deals with our study limitations (L356-364), and the second points to the contribution of our study to mountain tourism development literature (370-376).

Comment 3: managerial suggestions and recommendations are well done.

Response 3: We are so thankful for this comment.

Round 2

Reviewer 1 Report

No new suggestions